# `LIIR`: Learning Individual Intrinsic Reward in Multi-Agent Reinforcement Learning

**Yali Du**[*]
University College London
London, UK
yali.du@ucl.ac.uk

**Lei Han**[*]
Tencent AI Lab
Shenzhen, Guangdong, China
leihan.cs@gmail.com

**Meng Fang**
Tencent Robotics X
Shenzhen, Guangdong, China
mfang@tencent.com

**Tianhong Dai**
Imperial College London
London, UK
tianhong.dai15@imperial.ac.uk

**Ji Liu**
Kwai Inc.
Seattle, U.S.A.
ji.liu.uwisc@gmail.com

**Dacheng Tao**
UBTECH Sydney AI Centre,
The University of Sydney
NSW, Australia
dacheng.tao@sydney.edu.au

## Abstract

A great challenge in cooperative decentralized multi-agent reinforcement learning (MARL) is generating diversified behaviors for each individual agent when receiving only a team reward. Prior studies have paid many efforts on reward shaping or designing a centralized critic that can discriminatively credit the agents. In this paper, we propose to merge the two directions and learn each agent an intrinsic reward function which diversely stimulates the agents at each time step. Specifically, the intrinsic reward for a specific agent will be involved in computing a distinct proxy critic for the agent to direct the updating of its individual policy. Meanwhile, the parameterized intrinsic reward function will be updated towards maximizing the expected accumulated team reward from the environment so that the objective is consistent with the original MARL problem. The proposed method is referred to as learning individual intrinsic reward (`LIIR`) in MARL. We compare `LIIR` with a number of state-of-the-art MARL methods on battle games in StarCraft II. The results demonstrate the effectiveness of `LIIR`, and we show `LIIR` can assign each individual agent an insightful intrinsic reward per time step.

## 1 Introduction

Many real-world problems, such as traffic light control [1], coordination of autonomous vehicles [2], resources management [3] and multi-player video games [4, 5], can be naturally formulated into cooperative multi-agent systems, where the objective is to maximize the return in the perspective of a team of agents. When the agents are manipulated with a centralized controller which could access the joint or global state of all the agents, coordination among the agents is easier and the main effort of the controller is usually paid on finding an effective communication scheme among the agents. Examples include a wide range of approaches on designing effective centralized MARL architectures [5, 6, 7, 8].

---

[*]Equal contribution. Correspondence to the first two authors.

Unfortunately, when the agents are independently deployed and communications are disabled or prohibitive, each agent has to predict its own action conditioning on its partial observation trajectory. Without a centralized controller, each agent is responsible to collaborate with others on its own decision. This pushes much burden on the capability of each agent's policy. Worse still, in most of the real-world MARL applications, the agents only receive a team reward, from which it is difficult to deduce each agent's contribution to the team's success, making the learning and collaboration among agents nontrivial. There have been many efforts paid on distinguishing the agents' credit, known as the credit assignment problem in MARL [9, 10]. A general way is reward shaping [5, 11, 12], which, however, requires abundant human labor to assign precise rewards to each individual agent. Under some real-world tasks, such as reducing the latency in a traffic network, there might even not exist any clear choice of the reward functions for an individual agent (vehicle in the example). Another branch of commonly adopted methods try to design a centralized critic that is capable to distinguish the state-action values of the agents during training [9, 10], and then perform decentralized executions during testing.

Our approach builds a connection between reward shaping and critic learning. That is, we propose to learn each agent a parameterized individual intrinsic reward function by maximizing a centralized critic. The optimal intrinsic reward problem has been introduced in [13] for single agent learning scenarios and studied in some recent RL approaches [14, 15, 16]. Inspired by the concept, we propose to introduce the intrinsic reward design into multi-agent systems to distinguish the contributions of the agents when the environment only returns a team reward. Specifically, we learn each agent a parameterized intrinsic reward function that outputs an intrinsic reward for that agent at each time step to induce diversified behaviors. With these intrinsic rewards, we define each agent a distinct *proxy* expected discounted return that is a combination of the real team reward from the environment and the learned intrinsic reward. Using the actor-critic method, the individual policy of each agent is updated under the direction of the corresponding proxy critic. The parameters of the intrinsic reward functions are updated to maximize the standard accumulated discounted team return from the environment. Therefore, the objective of the entire procedure is consistent with that of the original MARL problem.

Insightfully, from an optimization perspective, the proposed method can be categorized to the bilevel optimization, where the problem of solving individual proxy objectives is nested within the outer optimization task which maximizes the standard multi-agent return. The parameters of the policy and the intrinsic reward function are treated as the parameters of the inner and outer optimization problems, respectively. We refer the proposed method to as learning individual intrinsic reward (LIIR) in MARL. Empirically, we show that LIIR outperforms a number of state-of-the-art MARL approaches on extensive settings in the battle game of StarCraft II. We also conduct insightful case studies to visualize the learned intrinsic reward, and the results demonstrate that the learned intrinsic reward function can generate diverse reward signals for the agents and the agents can also act diversely in a collaborative way.

## 2 Related Work

When considering a centralized controller in MARL, the controller takes the joint or global observation of the agents as input and outputs multiple actions for the agents in one step. Many studies have been proposed on pursuing effective communication architecture among the agents within a centralized controller. For example, densely connected communication layers or modules have been embedded in a centralized controller that directly outputs multi-head predictions for the agents [6, 5]. Recurrent neural networks (RNN) have also been introduced to enable a sequence of agents to communicate through the recurrent module [7]. However, in many MARL applications, the agents have to be separately deployed that each agent has to make its own decision conditioning on its partial observation.

Decentralized methods naturally deal with the above situation. The simplest approach is learning an individual policy or $Q$-function for each agent. This was first attempted with $Q$-learning [17], which was then extended with deep neural networks applied [18, 19]. Fully decentralized methods are limited under the case where only a team reward is given, since distinguishing the agents' contributions is difficult. To address the credit assignment problem in decentralized MARL, many existing methods utilize the framework with a centralized critic and decentralized policy. That is, the policies are deployed independently by taking individual observation as input, while the centralized

critic focuses on quantifying the differences among the agents. For example, the counterfactual multi-agent policy gradient [9] uses a counterfactual baseline to assign credits for the agents; the value decomposition network [20] decomposes the centralized value into a sum of individual agent values to discriminate their contributions; the QMIX [10] method adopts a similar idea that assumes the centralized $Q$-value function is monotonically increasing with the individual $Q$-values. Most of the existing methods focus on the architecture design of the critic, even strong assumptions on the value functions are unavoidable. Our method differs from these approaches that rather than working on the value functions, we propose to learn each agent an intrinsic reward at each time step. The benefits are that no assumptions are attached on the value functions and the agents are allocated an explicit immediate intrinsic reward at each time step to assign their credits.

Our work is also related to the optimal intrinsic reward design problem in single agent setting [21, 22, 23, 16, 24]. Some prior works have used heuristic metrics to define the intrinsic reward. For example, in [22] the intrinsic reward is defined as the squared difference between two consecutive states, and in [23] a metric named curiosity is used as the intrinsic reward. In [24] the learning of intrinsic reward is integrated with the update of the policy. A recent approach [16] proposes to parameterize the intrinsic reward function and alternatively updates the policy parameters and the intrinsic reward parameters. In this paper, we extend the setting to multi-agent system and use individual intrinsic reward function to distinguish the credits of the agents.

## 3    Background

### 3.1    Cooperative Multi-Agent Reinforcement Learning

We consider a fully cooperative multi-agent system, in which the agents need to be independently deployed without a central controller. The system can be described as a tuple as $\langle \mathcal{A}, S, U, P, r, \gamma, \rho_0 \rangle$. Let $\mathcal{A} = \{1, 2, \cdots, n\}$ denote the set of $n$ agents. Denote observation space of the agents as $S = \{S_1, S_2, \cdots, S_n\}$ and the action space of the agents as $U = \{U_1, U_2, \cdots, U_n\}$ respectively. At time step $t$, let $\mathbf{s}_t = \{s_t^i\}_{i=1}^n$ with each $s_t^i \in S_i$ being the partial observation from agent $i$. Accordingly, let $\mathbf{u}_t = \{u_t^i\}_{i=1}^n$ with each $u_t^i \in U_i$ indicating the action taken by the agent $i$. We overload notations and use $\mathbf{s}_t \in S$ to refer to the true state of the environment. $P(\mathbf{s}_{t+1}|\mathbf{s}_t, \mathbf{u}_t) : S \times U \times S \to [0, 1]$ is the state transition function. $r(\mathbf{s}_t, \mathbf{u}_t) : S \times U \to \mathbb{R}$ indicates the team reward function from the environment. In order to differentiate the team reward from the environment and the intrinsic reward that will be learned, we refer the team reward to as the extrinsic team reward $r^{\text{ex}}(\mathbf{s}_t, \mathbf{u}_t)$, following the usage in [16]. $\gamma \in [0, 1)$ is a discount factor and $\rho_0 : S \to \mathbb{R}$ is the distribution of the initial state $\mathbf{s}_0$. Let $\pi_i(u_t^i|s_t^i) : S_i \times U_i \to [0, 1]$ be a stochastic policy for agent $i$ and denote $\pi = \{\pi_i\}_{i=1}^n$. Let $J^{\text{ex}}(\pi) = \mathbb{E}_{\mathbf{s}_0, \mathbf{u}_0, \cdots}[R_0^{\text{ex}}]$ with $R_t^{\text{ex}} = \sum_{l=0}^{\infty} \gamma^l r_{t+l}^{\text{ex}}$ denoting the expected discounted extrinsic reward, where $\mathbf{s}_0 \sim \rho_0(\mathbf{s}_0)$, $u_t^i \sim \pi_i(u_t^i|s_t^i)$ for $i \in \mathcal{A}$, and $\mathbf{s}_{t+1} \sim P(\mathbf{s}_{t+1}|\mathbf{s}_t, \mathbf{u}_t)$. Define the extrinsic value function as $V_\pi^{\text{ex}}(\mathbf{s}_t) = \mathbb{E}_{\mathbf{u}_t, \mathbf{s}_{t+1}, \cdots}[R_t^{\text{ex}}]$. We aim to find optimal policies $\pi^* = \{\pi_i^*\}_{i=1}^n$ that achieve the maximum expected extrinsic team reward $J^{\text{ex}}(\pi^*)$.

### 3.2    Centralized Learning with Decentralized Execution

Centralized learning with decentralized execution (CLDE) is a commonly used architecture to learn a centralized critic to update the decentralized policies during training. In CLDE, actor-critic (AC) style methods [25, 26, 27, 28, 29] are often selected. In our case, AC algorithms use $n$ independent parameterized policies $\pi_{\theta_i}$ for $i \in \mathcal{A}$ and update $\theta_i$ by maximizing the expected extrinsic reward $J^{\text{ex}}(\theta_1, \theta_2, \cdots, \theta_n) = \mathbb{E}_{\mathbf{s}, \mathbf{u}}[R^{\text{ex}}]$ using the policy gradient

$$\nabla_{\theta_i} J^{\text{ex}}(\theta_1, \theta_2, \cdots, \theta_n) = \mathbb{E}_{\mathbf{s}, \mathbf{u}}\left[\nabla_{\theta_i} \log \pi_{\theta_i}(u_i|s_i) A_\pi(\mathbf{s}, \mathbf{u})\right], \tag{1}$$

where $A_\pi(\mathbf{s}, \mathbf{u})$ is the centralized critic. There are several ways to estimate $A_\pi(s, \mathbf{u})$. For example, $A_\pi(\mathbf{s}, \mathbf{u}) = r^{\text{ex}}(\mathbf{s}, \mathbf{u}) + V^{\text{ex}}(\mathbf{s}') - V^{\text{ex}}(\mathbf{s})$ is the standard advantage function [27, 28], where $\mathbf{s}'$ is the successive state of the agents. In [9], $A_\pi(s, \mathbf{u})$ is defined as an estimated state-action value function minus a counterfactual baseline.

### 3.3    Parameterized Intrinsic Reward

A recent study [16] has investigated learning a parameterized intrinsic reward function in single agent setting. The idea is to explicitly define the intrinsic reward function as $r_\eta^{\text{in}}(s, a)$ for a state-action pair

$(s, a)$ of the agent, and it is summed up with the extrinsic reward $r^{\text{ex}}(s, a)$ from the environment to serve as the return signal for updating the policy. The intrinsic reward parameter $\eta$ is updated towards maximizing the expected extrinsic reward $J^{\text{ex}}$. The intuition for updating $\eta$ is to find the effect that the change on $\eta$ would influence the extrinsic value through the change in the policy parameters. This technique can be viewed as an instance of meta learning [30, 31, 32]; the intrinsic reward function serves as a meta-learner that learns to improve the agents objective. In our case, we extend the intrinsic reward learning method to deal with decentralized MARL problem and we use the intrinsic rewards to diversely stimulate the agents to learn from the environment.

# 4  Method

In this section, we formally propose the LIIR method. We first provide a formal definition of the considered problem based on what have been introduced in Section 3, then we introduce a bilevel optimization algorithm for solving the proposed objective.

## 4.1  The Objective

By defining an intrinsic reward function $r^{\text{in}}_{\eta_i}(s_i, u_i)$ which is parameterized by $\eta_i$ and takes a state-action pair $(s_i, u_i)$ of an individual agent $i$ as input, we propose to assign agent $i$ a distinct *proxy* reward

$$r^{\text{proxy}}_{i,t} = r^{\text{ex}}_t + \lambda r^{\text{in}}_{i,t}, \tag{2}$$

at time step $t$. In (2), we have omitted the arguments of the reward functions for simplicity, and $\lambda$ is a hyper-parameter that balances the extrinsic team reward and the distinct intrinsic reward. Note that in the standard MARL problem with a team reward, there does not exist any distinct reward for each agent. Now, after creating each agent a proxy reward $r^{\text{proxy}}_{i,t}$ at time step $t$, we accordingly define a discounted proxy reward for each agent $i$ as

$$R^{\text{proxy}}_{i,t} = \sum_{l=0}^{\infty} \gamma^l (r^{\text{ex}}_{t+l} + \lambda r^{\text{in}}_{i,t+l}), \tag{3}$$

and the proxy value function for agent $i$ as

$$V^{\text{proxy}}_i(s_{i,t}) = \mathbb{E}_{u_{i,t}, s_{i,t+1}, \cdots}[R^{\text{proxy}}_{i,t}]. \tag{4}$$

Different from the extrinsic (standard) value $V^{\text{ex}}$, these proxy value functions $V^{\text{proxy}}_i$'s do not have any physical meanings and they will be only used for updating the individual policy parameters $\theta_i$'s. Now, the considered overall objective is defined as

$$\max_{\boldsymbol{\eta}, \boldsymbol{\theta}} \quad J^{\text{ex}}(\boldsymbol{\eta}), \tag{5}$$
$$\text{s.t.} \quad \theta_i = \arg\max_{\theta} J^{\text{proxy}}_i(\theta, \boldsymbol{\eta}), \quad \forall i \in [1, 2, \cdots, n]$$

where $J^{\text{proxy}}_i := \mathbb{E}_{s_{i,0}, u_{i,0}, \cdots}\left[R^{\text{proxy}}_{i,0}\right]$ depending on $\theta_i$ and $\boldsymbol{\eta}$, $\boldsymbol{\eta}$ indicates the intrinsic reward parameter set $\{\eta_1, \eta_2, \cdots, \eta_n\}$ and $\boldsymbol{\theta}$ indicates the policy parameter set $\{\theta_1, \theta_2, \cdots, \theta_n\}$.

In problem (5), the goal is to maximize $J^{\text{ex}}$ through optimizing $\boldsymbol{\eta}$, while the policy parameter $\theta_i$ is optimized by maximizing the proxy expected discounted return $J^{\text{proxy}}_i$ for agent $i$. The advantage is that by learning a distinct intrinsic reward for each agent per time step, the agents will be diversely stimulated and this will accumulatively influence the policy learning via the policy gradient. Moreover, from an optimization perspective, problem (5) can be viewed as a bilevel optimization problem, since the problem of maximizing the individual proxy expected returns is nested within the outer optimization task, which is maximizing the extrinsic expected return. In the next subsection, we will discuss how $J^{\text{ex}}$ is connected with the intrinsic reward parameter $\boldsymbol{\eta}$.

## 4.2  Algorithm

As a bilevel optimization problem, at each iteration, the policy parameters are updated with respect to the inner proxy tasks, while the intrinsic reward parameters are updated to maximize the extrinsic expected return.

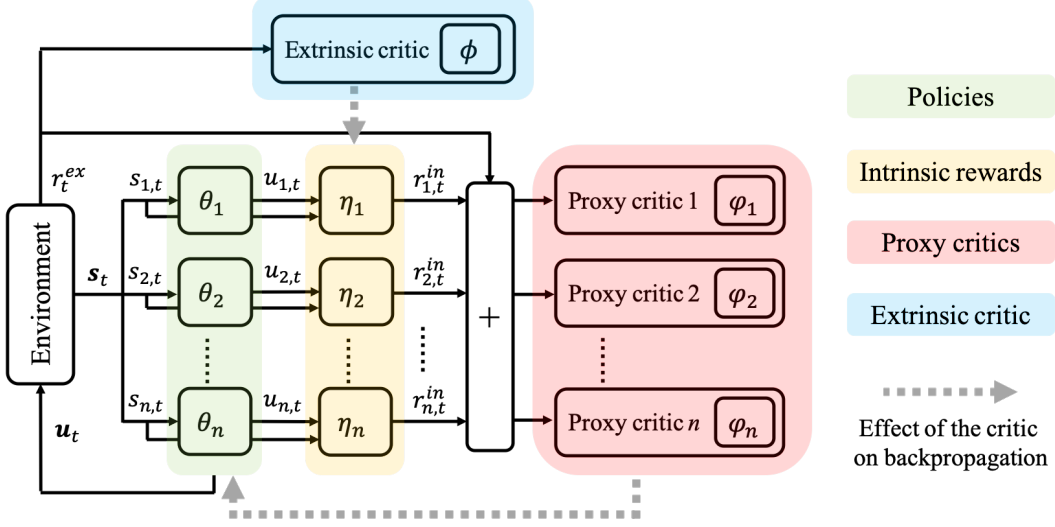

Figure 1: Architecture of the `LIIR` method. The architecture contains four parameter components: $\theta_i$'s for policies, $\eta_i$'s for intrinsic reward, and $\varphi_i$'s and $\phi$'s for extrinsic and proxy values respectively.

Specifically, the policy parameter of each agent is updated by the policy gradient with its proxy critic. Given a trajectory generated by the policy $\pi_{\theta_i}$, $\theta_i$ can be updated by applying the policy gradient defined in (1):

$$\nabla_{\theta_i} \log \pi_{\theta_i}(u_i|s_i) A_i^{\text{proxy}}(s_i, u_i), \tag{6}$$

where $A_i^{\text{proxy}}(s_i, u_i)$ is the proxy critic that can be chosen in a variety of ways [25, 26, 27, 28]. For example, $A_i^{\text{proxy}}(s_i, u_i) = R_i^{\text{proxy}}$ leads to the REINFORCE algorithm [26]. In this paper, we choose $A_i^{\text{proxy}}(s_i, u_i) = r_i^{\text{proxy}}(s_i, u_i) + V_{\varphi_i}^{\text{proxy}}(s_i') - V_{\varphi_i}^{\text{proxy}}(s_i)$ as the advantage function [27, 28], where $V_{\varphi_i}^{\text{proxy}}$ is the proxy value parameterized by $\varphi_i$ and $s_i'$ is the next state of agent $i$ in the trajectory. Given (6) and a policy learning rate $\alpha$, the updated policy parameter $\theta_i'$ can be represented as $\theta_i' = \theta_i + \alpha \nabla_{\theta_i} \log \pi_{\theta_i}(u_i|s_i) A_i^{\text{proxy}}(s_i, u_i)$.

Then, we build the connection between $\boldsymbol{\eta}$ and $J^{\text{ex}}$ and specify the updating procedure for $\boldsymbol{\eta}$. Given the updated policy parameters $\theta_i'$'s, using the chain rule, we have

$$\nabla_{\eta_i} J^{\text{ex}} = \nabla_{\theta_i'} J^{\text{ex}} \nabla_{\eta_i} \theta_i'. \tag{7}$$

The spirit of (7) is to formulate the effect of the change of $\eta_i$ on influencing $J^{\text{ex}}$ through its influence in the updated policy parameter $\theta_i'$. This is a commonly adopted technique in meta-gradient learning [30, 31, 32, 33]. Computing the meta-gradient $\nabla_{\eta_i} J^{\text{ex}}$ requires new samples generated by the updated policy parameter $\theta_i'$, while this can be avoid by reusing the samples generated by $\theta_i$ with importance sampling [16]. In (7), $\nabla_{\theta_i'} J^{\text{ex}}$ can be estimated by stochastic gradient as

$$\nabla_{\theta_i'} \log \pi_{\theta_i'}(u_i|s_i) A^{\text{ex}}(\boldsymbol{s}, \boldsymbol{u}), \tag{8}$$

where $A^{\text{ex}}(\boldsymbol{s}, \boldsymbol{u})$ is the centralized extrinsic critic. Similar to proxy critics, we choose $A^{\text{ex}}(\boldsymbol{s}, \boldsymbol{u}) = r^{\text{ex}}(\boldsymbol{s}, \boldsymbol{u}) + V_\phi^{\text{ex}}(\boldsymbol{s}') - V_\phi^{\text{ex}}(\boldsymbol{s})$, where $V_\phi^{\text{ex}}(\boldsymbol{s})$ is the extrinsic value parameterized by $\phi$. The second term in (7) can be derived as

$$\begin{aligned}
\nabla_{\eta_i} \theta_i' &= \nabla_{\eta_i}[\theta_i + \alpha \nabla_{\theta_i} \log \pi_{\theta_i}(u_i|s_i) A_i^{\text{proxy}}(s_i, u_i)] \\
&= \alpha \lambda \nabla_{\theta_i} \log \pi_{\theta_i}(a_i|s_i) \nabla_{\eta_i} r_i^{\text{proxy}}(s_i, u_i).
\end{aligned} \tag{9}$$

Fig. 1 gives an illustration of the entire architecture of the `LIIR` method. A sketch of the optimization algorithm is presented in Algorithm 1.

## 5  Experiments

In this section, we first evaluate `LIIR` on a simple 1D pursuit game specifically designed for the considered settings to see whether `LIIR` can learn reasonable distinct intrinsic rewards. Then, we

**Algorithm 1** The optimization algorithm for `LIIR`.

---
**Input:** policy learning rate $\alpha$ and intrinsic reward learning rate $\beta$.
**Output:** policy parameters $\boldsymbol{\theta}$ and intrinsic reward parameters $\boldsymbol{\eta}$.

1: **Init**: initialize $\boldsymbol{\theta}$ and $\boldsymbol{\eta}$;
2: **while** termination is not reached **do**
3:     Sample a trajectory $\mathcal{D} = \{\mathbf{s}_0, \mathbf{u}_0, \mathbf{s}_1, \mathbf{u}_1, \cdots\}$ by executing actions with the decentralized policies $\{\pi_{\theta_1}, \cdots, \pi_{\theta_n}\}$;
4:     Update $\boldsymbol{\theta}$ according to (6) with learning rate $\alpha$;
5:     Compute (8) using new samples from $\{\pi_{\theta'_1}, \pi_{\theta'_2}, \cdots, \pi_{\theta'_n}\}$ or reuse $\mathcal{D}$ to replace (8) with $\frac{\nabla_{\theta'_i} \pi_{\theta'_i}(u_i|s_i)}{\pi_{\theta_i}(u_i|s_i)} A^{\text{ex}}(\boldsymbol{s}, \boldsymbol{u})$;
6:     Update $\boldsymbol{\eta}$ according to (7), step 5 and (9) with learning rate $\beta$;
7: **end while**

---

comprehensively study `LIIR` in several challenging micromanagement games in the game of StarCraft II, and compare `LIIR` with a number of state-of-the-art MARL methods.[2]

## 5.1 A Simple 1D Pursuit Study

We design a simple game named *1D Pursuit* to provide a fast verification for the quality of the intrinsic reward learned by `LIIR`. In *1D pursuit*, a team of two agents are initially assigned with some random integers denoted by $x$ and $y$ respectively, and each agent could take actions from $\{+1, -1, 0\}$ to either increase, decrease or keep its value to approach a target value $z$ that is unknown to the agents. For a collaborative setting, the team reward for the two agents is set to be inversely proportional to the summation of their absolute differences between their values and the target value. That is, both the two agents should adjust their values towards the target value. The observation of each agent is a two-dimension vector containing its current integer value and another agent's integer value.

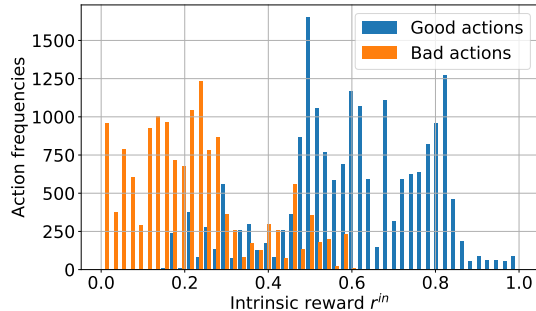

Figure 2: The distribution of the learned intrinsic rewards v.s. frequencies (counts) of taking "Good" and "Bad" actions from 1000 1D pursuit games.

The team reward is set to be $+0.01$ if both agents take actions that approaching the target value, $-0.01$ if both agents take actions that moving away from the target value, and $0$ otherwise. The target value is set to be $0$. The initial integers for the two agents are randomly generated from $\{-10, ..., 10\}$.

We implement `LIIR` based on the architecture depicted in Fig. 1. The detailed network structure is provided in the supplementary material. In Fig. 2, we plot the histogram of the distributions of the intrinsic reward averaged from 1000 episodes. We denote actions approaching the target as "Good" actions and actions moving away from the target as "Bad" actions. The result shows that `LIIR` can assign reasonable intrinsic reward to the agents.

## 5.2 StarCraft II Micromanagement

In this subsection, we comprehensively evaluate the proposed `LIIR` method in the game of StarCraft II based on the learning environment SC2LE [34] and mini-game settings in SMAC [35]. We compare the `LIIR` method with a number of state-of-the-art MARL methods that use the CLDE architecture. We also provide some insightful case studies to visualize the learned intrinsic rewards.

StarCraft II is a popular real-time strategy game and it has been studied under MARL settings [9, 10, 7, 36, 37]. In the experiments, we consider symmetric battle games in StarCraft II , where both single type agents and mixed type agents are considered.

Specifically, the considered scenarios contain 3 Marines vs. 3 Marines (3M), 8 Marines vs. 8 Marines (8M), 2 Stalkers & 3 Zealots vs. 2 Stalkers & 3 Zealots (2S3Z), and 3 Stalkers & 5 Zealots vs. 3

Stalkers & 5 Zealots (3S5Z). In these settings, Marine and Stalker are units of Terran and Protoss, respectively, and both of them can attack enemies at a distance, while Zealot is a melee unit of Protoss and it can only attack enemies who stand close to it. In all these games, only the units from self side are treated as agents.

Each agent is described by several attributes including the health point (HP), weapon cooling down (CD), shield (for 2S3Z and 3S5Z), unit type, last action and the relative distance of the observed units. The enemy unit is described in the same way except that CD is excluded. The partial observation of an agent is composed by the attributes of the units, including both the agents and the enemy units, shown up within its view range that is a circle with a certain radius. The action space contains 4 move directions, $k$ attack actions where $k$ is the fixed maximum number of the enemy units in a map, stop and none-operation. The input dimension and the output action dimension are fixed with a certain ordering over the agents and enemy units. Dead enemy units will be masked out from the action space to ensure the executed action is valid. At each time step, the agents receive a joint team reward which is defined by the total damage of the agents and the total damage from the enemy side. In all the scenarios, following the configurations in [9, 10], we train the agents against the build-in AI opponent. More detailed settings can be acquired from the SMAC environment [35].

### 5.2.1 Compared Methods and Training Details

The considered methods for evaluation include

- independent $Q$-learning (IQL) [17]: IQL trains decentralized $Q$-functions for each agent. Since the observation and action spaces of the agents are the same within a specific environmental setting, a policy will be shared across all the agents;

- independent actor-critic (IAC) [9]: IAC is similar to IQL except that it adopts the actor-critic method;

- Central-V [9]: the method learns a centralized critic with decentralized policies. Similarly, all agents share the same policy network;

- COMA [9]: the method learns a centralized critic that is the state-action value minus a counterfactual baseline;

- QMIX [10]: the method learns decentralized $Q$-function for each agent with the assumption that the centralized $Q$-value is monotonically increasing with the individual $Q$-values. In the implementations, the agents share the same $Q$-function;

- LIIR: the proposed method. In the experiments, the agents share the same policy, intrinsic reward function and proxy critic. Since each agent has its own partial observation, sharing policy parameters does not imply that they act the same.

For COMA and QMIX, we use their original implementations, in which the main policy network or $Q$-network consist of some fully connected (FC) layers and a GRU module.[3] All the other methods adopt similar network structures compared to COMA and QMIX. As depicted in Fig. 1, the parameters of `LIIR` contain 4 components corresponding to the shared policy parameter $\theta$, intrinsic reward parameter $\eta$, proxy value parameter $\varphi$ and extrinsic value parameter $\phi$. To achieve fair comparison, we set the policy network structure, i.e., $\theta$, as what is exactly used for COMA's policy network. Then, we compress the other parameters $\eta$, $\varphi$ and $\phi$ to let their total size equal to the parameter size of the remaining part in COMA. More details can be found in the supplementary material. All the methods are trained with 3 millions of steps in 3M and 8M, and with 10 millions of steps for 2S3Z and 3S5Z. The hyper-parameter $\lambda$ in (2) is set to 0.01 throughout the experiments (we tried different choices of $\lambda$ while we found that the results did not differ much). We use a fixed learning rate of 5e-4 and use batches of 32 episodes for all the methods. We use 32 actors to generate the trajectories in parallel, and use one NVIDIA Tesla M40 GPU for training.

### 5.2.2 Results

To evaluate the performance of each method, we freeze the training every 100 episodes and test the model over 20 episodes to compute an average test winning rate. The entire training procedure is

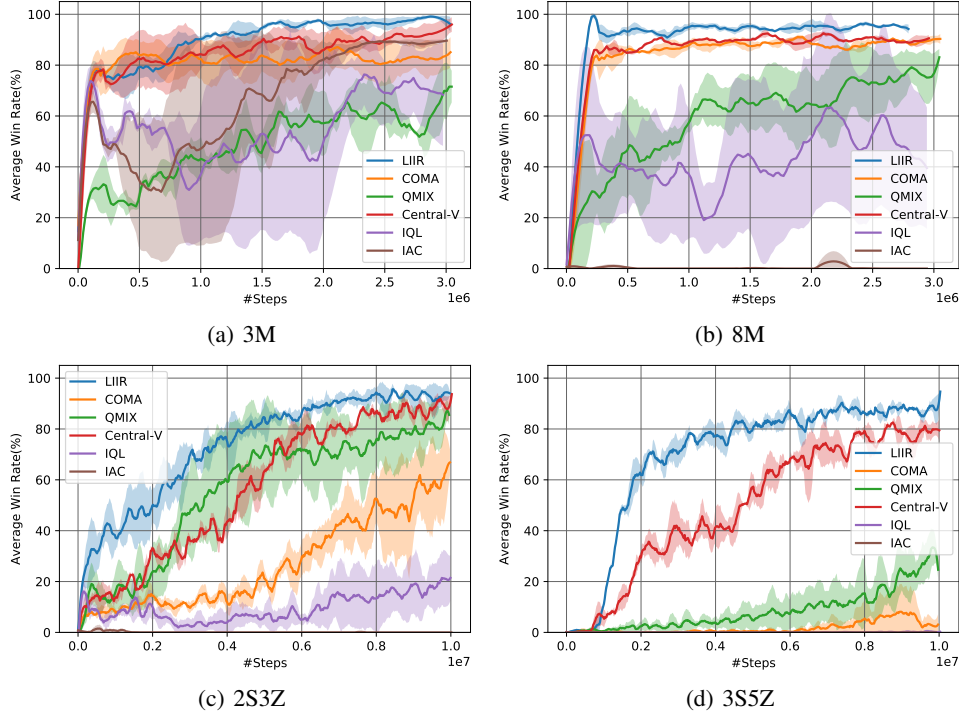

Figure 3: Test winning rates vs. training steps of various methods on all the scenarios.

repeated for 5 times to plot the winning rate curve with standard deviation. The results are reported in Fig. 3, where the averaged winning rates vs. the training steps on all the battle scenarios are given.

In 3M which is the simplest game, all the test winning rates keep increasing as the training steps increase. In 8M, 2S3Z and 3S5Z, the independent learning methods, i.e., IQL and IAC, fail to learn a good policy for the agents and the methods using a CLDE architecture always outperform the independent learning methods. In 3M and 8M, COMA and Central-V show comparable performance, while in 2S3Z and 3S5Z, Central-V outperforms QMIX and COMA. For all these scenarios, the LIIR method consistently shows the best performance, and it achieves around 90% winning rate in all the scenarios. This demonstrates that learning the intrinsic reward function can ultimately induce better trained policies.

### 5.2.3 Visualizing the Learned Intrinsic Reward

In addition to evaluate the performance of the trained policy in Section 5.2.2, we are more curious about how much effect the learned intrinsic reward function actually contributes to the policy learning. In order to figure out what has been learned in the intrinsic reward function, we propose to explicitly visualize these rewards. That is, we plot the learned intrinsic reward of each agent at each time step in a complete trajectory during testing. It is worth mentioning that during testing the intrinsic rewards are independent with the learned policy, and these rewards will not be used at all when generating the trajectory. For clarity, we randomly choose two test replays in 3M and 2S3Z which contain fewer agents to plot all the agents' intrinsic rewards. Figs. 4 and 5 show the intrinsic rewards in 3M and 2S3Z, respectively. We also attach some auxiliary snapshots to explain some interesting segments in the curves. In all the snapshots, the red colored units indicate the agents controlled by LIIR.

In Fig. 4(a), agent 1 is dead at time step 9, and we can observe that its intrinsic reward turns to be very low after time step 6 compared to the other two agents. As revealed by Figs. 4(b) and (c), at time step 6, all the three agents focus fire on one of the enemy Marine, while agent 1 has the lowest HP; after that, agent 1 still keeps firing instead of running away from the enemies and the intrinsic reward function predicts a low $r_1^{in}$, indicating that $u_1 = \texttt{attack}$ is not a good action at that time; finally, agent 1 dies at time step 9 and the corresponding intrinsic reward is very low.

In Fig. 5(a), after time step 27, we see that agent 2's intrinsic reward increases a lot compared to the other agents. Figs. 5(b) and (c) provides a clear explanation that at time step 27, agent 2 (with

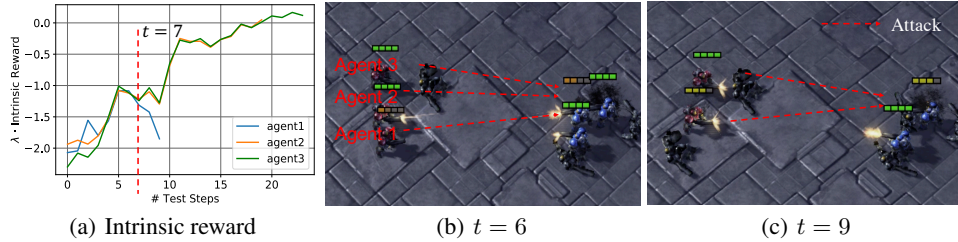

(a) Intrinsic reward        (b) $t = 6$        (c) $t = 9$

Figure 4: An example of the intrinsic reward curves and auxiliary snapshots on 3M.

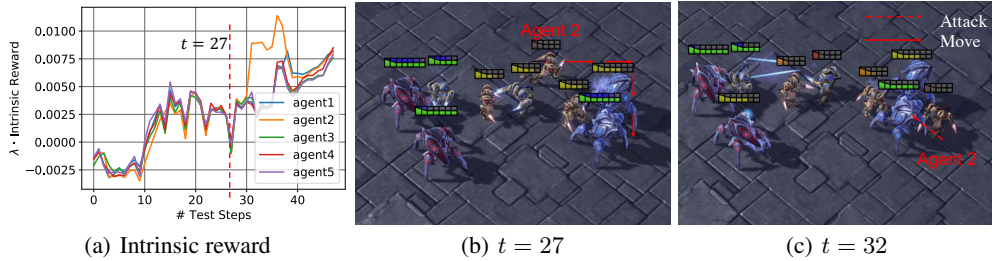

(a) Intrinsic reward        (b) $t = 27$        (c) $t = 32$

Figure 5: An example of the intrinsic reward curves and auxiliary snapshots on 2S3Z.

low HP) stops firing and runs along the red arrows (the move actions only take 4 directions here) to avoid the attack from the enemy Zealot; until reaching an enemy Stalker at time step 32, agent 2 starts attacking the Stalker which is finally killed. Moreover, the overall trend of both the curves in Figs. 4(a) and 5(a) keeps increasing, indicating that the controlled team finally wins the game.

Besides visualizing the two episodes illustrated above, we also provide overall statistics of the learned intrinsic reward. We collect the intrinsic reward for the action "attack" when the corresponding health points are lower than 50% from 100 test episodes. We then compute the cosine similarity (a value in [-1, 1]) between the health point and the intrinsic reward. The averaged cosine similarity is 0.55 for 2S3Z and 0.67 for 3M. The results show that the health point and intrinsic reward are positively correlated. That is, when the health point is low, the intrinsic reward is generally low for taking the "attack" action as well, which is reasonable in this scenario.

The above case studies demonstrate that the learned intrinsic reward can indeed provide diverse feedback signals for the agents and these signals are very informative in evaluating the agents' immediate behaviors.

## 6 Conclusion

We have proposed a novel multi-agent reinforcement learning algorithm, which learns an individual intrinsic reward for each agent. The method can assign each agent a distinct intrinsic reward so that the agents are stimulated differently, even when the environment only feedbacks a team reward. Given the intrinsic reward for each agent, we define each of them a proxy critic to direct their policy learning via actor-critic algorithms. We show that the formulated multi-agent learning problem can be viewed as a bilevel optimization problem. Our empirical results carried on the battle games in StarCraft II demonstrate that learning the intrinsic reward function could eventually induce better trained policy compared with a number of state-of-the-art competitors. We further perform two case studies to visualize the learned intrinsic reward values, and the results provide clear explanations on the effects of the learned intrinsic rewards.

For future work, we are interested in applying the `LIIR` method to more challenging scenarios, such as real-world traffic control with many agents and competitive multi-agent systems. Moreover, in addition to the simple summation form in (2), it is also interesting to investigate the optimal form of the proxy reward function.

**Acknowledgments**

The authors would like to thank anonymous reviewers for their constructive comments. Yali Du is during an internship at Tencent AI Lab when working on this project.

## Footnotes

[2]The source codes of `LIIR` are available through `https://github.com/yalidu/liir`.

[3]`https://github.com/oxwhirl/pymarl`

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
