[Supplementary Material]

# `LIIR`: Learning Individual Intrinsic Reward in Multi-Agent Reinforcement Learning – Supplementary Material

## 1   Detailed Network Structures for `LIIR` Used in The Experiments

The neural network structures used for the StarCraft II Micromanagement games are depicted in Fig. 1. In Fig. 1(a), the network for the proxy value $v_i^{proxy}$ and the intrinsic reward $r_i^{in}$ is a fully connected neural net with two hidden layers of dimension 128, which are shared by all agents. In Fig. 1(b), the extrinsic value function uses a similar fully connected neural net which takes a joint observation of all agents as input and outputs the value for the entire team. The policy $\pi_i$ is a recurrent neural network with a GRU unit and is shared among all agents, as depicted in Fig. 1(c).

For 1D pursuit, the networks share exactly the same structure as described above with only reduced dimensions for the hidden layers, as shown in Fig. 2

(a) Proxy network

(b) Extrinsic critic

(c) Actor network

Figure 1: Neural network structures for Starcraft II.

(a) Proxy network

(b) Extrinsic critic

(c) Actor network

Figure 2: Neural network structures for *1D Pursuit*.