[Reviews · NeurIPS 2019]

Reviewer 1



The paper is well written. I do not have any clarity issues. To a large extent, the paper is a successor of the work by Zeyu et al. [16]: it is a straightforward extension of learning intrinsic reward to the cooperative multi-agent setting. Therefore, the technical contributions are somewhat limited. PS: Another paper on multi-agent intrinsic reward. Liu, Bingyao, Satinder Singh, Richard L. Lewis, and Shiyin Qin. "Optimal rewards for cooperative agents." I have given my list of three most significant contributions and suggestions for improvement in other sections of the review. Here I have some minor questions: Any particular reason why the authors did not choose all the tasks used in the COMA paper, for the purpose of comparison? In the COMA paper [9], the tasks are 3M, 5M, 5W, and 2D3Z. In this paper, we have 3M, 8M, 2S3Z, 3S5Z.

Reviewer 2



Originality: The ideas introduced here are certainly not new, but extending intrinsic rewards to multi-agent RL settings is for sure an interesting research avenue, especially when one is interested in decentralized multi-agent settings were no communication is possible between agents. Quality: The paper is well written. Clarity: It is a fairly convoluted method, with many components. A better overview of the algorithm could be useful (perhaps in supplementary materials). Furthermore, not all the details regarding the experimental setup and parameter choice is specified. This information is important for reproducibility reasons, and could also be included in supplementary materials. A few examples include the beta learning rate, was there any parameter search performed for lambda, more in-depth view of the networks' architectures. Significance: The method is compared against state-of-the-art methods and does show improvements in the selected scenarios. The authors also perform a short analysis of what intrinsic reward the agents learn and how it affect their behaviour. ------------------------------------ Post-rebuttal: I appreciate the authors efforts to answer the raised concerns and I think the additional experiments, analysis and explanations will improve the work. I will maintain my score, given the novelty level of the work.

Reviewer 3



This work deals with learning individual intrinsic rewards (IR) for muti-agent RL (MARL). Overall, the method provided is a straightforward application of a known IR method to MARL, the results are promising and the writing is clear. As such, this work has limited novelty but provides good empirical contributions, though these too could be improved by considering more domains. A more detailed review of the paper, along with feedback and clarifications required are provided below. The work is motivated by the claim that providing individual IRs to different agents in a population (in a MARL setting) will allow diverse behaviours. * Is it not possible that the IR policies learnt all look similar and the thus the behaviour that emerges is similar? The analysis at the end of the paper shows that a lot of the learned IR curves do overlap. Please provide more justification for this motivation. The work clearly describes related work and how the approach here differs. The main contribution is to apply the meta-gradient based approach in “On learning intrinsic rewards for policy gradient methods” ([16] as per the paper) to the multi-agent setting. * This looks to be a straightforward application where each agent has the LIRPG approach applied. Please provide succinct details of any modifications that are required to apply this and any differences in implementation. The method section can be shortened, as most of the algorithm and objective are the same as the original LIRPG algorithm uses. A range of methods are compared to the in the experimental section: independent q-learning/actor-critic, central critics, counterfactual critics, QMIX and the proposed approach (LIIR). * Please clarify what is meant by “share the same policy”: do they share the same policy network weights or also the exact same policy output? Do all agents get the same observation? If so, what is the difference between IAC and central-v? Is the only change how the V is updated, whereas policy is the same? * How is the parameter \lambda tuned for the agent? Lastly, the result sections show clear benefits of this approach. This method, along with several baseline is applied to a set of mini games for Starcraft. The analysis is promising and show that the method learns an interesting policy that captures the dynamics of the game. Overall this is a good contribution but for an empirical paper this could be strengthened by considering more domains or tasks and demonstrating the ability of this method to work across the board.

[Author Response · NeurIPS 2019]

We thank the reviewers for all of these valuable comments. We provide point by point responses below.

**To Reviewer #1**

**Q1: About the significance and contribution. A:** The technical part of LIIR is indeed inspired by what was proposed
in [16], while we think the considered problem and the definitions of the individual intrinsic reward and proxy value are
new for MARL research and the reported results could contribute to the MARL domain. Moreover, we think formulating
the intrinsic reward learning problem into bi-level optimization is new from the perspective of meta-gradient learning,
which is the key to make the extension to the multi-agent case natural.

**Q2: "Another paper...'Optimal rewards for cooperative agents'..." A:** We have carefully read the paper and we
think the method differs from ours that it does not consider the centralized learning and decentralized execution
architecture and the learning of its intrinsic reward is integrated with the update of the policy while we cast the intrinsic
reward learning as a meta-gradient learning problem. We will provide more discussions in the revision.

**Q3: "...why the authors did not choose all the tasks used in the COMA paper..." A:** We think 8M, 2S3Z and 3S5Z
are more challenging tasks compared to 5M, 5W and 2D3Z. We also noticed that 2S3Z and 3S5Z were studied in QMIX
[33] which had been demonstrated to be superior than COMA. Therefore, we chose a mixture of the scenarios of these
used in COMA and those used in QMIX. Actually, all these settings are based on the SMAC framework.

**Q4: "...deeper analyses of the learned intrinsic reward..." A:** Thanks for the comments. According to your sug-
gestion, we have collected all the $r_{in}(s, a)$'s for the action $a =$'attack' when the corresponding HP's are lower than
a percent of 50% from 100 test episodes, and we compute their cosine similarity coefficient (a value in [-1, 1]). The
averaged cosine similarity is 0.55 for 2S3Z and 0.67 for 3M, showing that when the HP is low, the intrinsic reward
generally shows a low value for taking 'attack' action as well. We will include these discussions in the revision.

**Q5: "...more analysis/explanation...more convincing results, maybe in other domains." A:** Thanks for the com-
ments. We think 3S5Z is the most complicated task among the four settings, and in 3S5Z agents might act more
diversely and hence LIIR could perform much better. We will perform more explanations for these experimental results.
For other domains, per the suggestion, we have designed a new game named *1D Pursuit* to provide a fast evaluation of
the generality of LIIR. In *1D pursuit*, two agents are assigned with two initial integers $x$ and $y$, and the agents could
take actions from $\{+1, -1, 0\}$ to increase, decrease or keep their values to approach a target value $z$. The team reward
is set to be inversely proportional to $|z - x| + |z - y|$. We find that LIIR could easily assign a reasonable intrinsic
reward for each agent. Specifically, we denote actions approaching (moving away from) the target as "good" ("bad")
actions, and we plot the histogram of the intrinsic reward distributions from 100 episodes in Fig. 1 (d). The figure shows
that LIIR can learn reasonable intrinsic reward for the agents. So we think LIIR is a general approach.

**Q6: "Is the idea well suited for the competitive scenarios?" A:** Applying LIIR to competitive MARL scenarios is
very interesting and it should not be a complicated extension. For example, under competitive settings, there should
also exists a global score measuring the game status of all the agents, and one can design an intrinsic reward function
for each of these competitive agents to differentiate their gains (which might not be symmetric). We are interested to
investigate such scenarios in the future work.

**To Reviewer #2**

**Q1: "The readability and reproducibility can certainly be improved..."**
**A:** Thanks for your comments. We followed the parameter settings in COMA
and QMIX, so we omitted some details on describing the experiment. We
will enrich these information in the revision. Specifically, we fix the learning
rates $\alpha$ and $\beta$ to be $5 \times 10^{-4}$ in all experiments. Following [16], $\lambda$ is set to be
0.01. We set the batch size as 32. An overview of the network architecture
is shown in Fig. 1. Codes will be released for reproducing all the results.

**To Reviewer #3**

**Q1: "...the learned IR curves do overlap..." A:** Most of the agents might
have similar observations so their intrinsic rewards are similar, while we
have performed more analyses of the learned intrinsic reward. Please refer to the response to Q4 of Reviewer #1.

Figure 1: Network architecture and new results on the 1D pursuit game.

**Q2: "...straightforward application...any differences in implementation..." A:** In the considered MARL problem,
we have to define each agent an individual intrinsic reward. An important difference compared to [16] is that for MARL
problem the original objective is maximizing over the expected extrinsic team return, and a direct connection between
the extrinsic team return and the individual intrinsic reward functions is not straightforward. In LIIR, we build proxy
value functions for the agents and connect them with the team return via the bi-level optimization problem.

**Q3:"...what is meant by 'share the same policy'...How is $\lambda$ tuned..." A:** "share the same policy" indicates the agents
share the same policy parameters. Each agent may have different partial observations, so the output actions are also
different. In decentralized settings, IAC differs from Central-V that their value functions are distinctly defined and
Central-V uses a centralized critic while IAC uses independent critics. Following [16], the parameter $\lambda$ is set as 0.01.
We indeed tried different choices of $\lambda$ while we found that the results did not differ much.

**Q4: "...could be strengthened by considering more domains..." A:** Thanks for the suggestion. We have studied
another task for evaluating the generality of LIIR. Please refer to the response to Q5 of Reviewer #1.

[Meta-Review · NeurIPS 2019]

The paper extends the idea of learning intrinsic rewards to the centralized learning - decentralized execution, cooperative multi-agent setting. This setting has become popular in past years, as a setting that has high potential for real world applications and being amenable to progress towards tractable solutions. The approach presented by this work is easy to conceptually simple and well motivated. The authors empirically show that it outperforms existing state of the art approaches on challenging StarCraft benchmark tasks. Reviewers raised several concerns about the paper, including clarity (experiment details, precise description of the approach and distinction from existing approaches), and the need for further analysis. Most of the noted points were addressed by the authors in the rebuttal. In particular, the authors provided an additional diagnostic experiment that sheds light on how individual agents learn different internal proxy rewards. Overall, the paper is assessed as making a valuable contribution to NeurIPS. The proposed approach is well motivated and works well in practice. I strongly urge the authors to carefully consider all reviewer feedback when preparing the camera ready version.